# A Private 16q24.2q24.3 Microduplication in a Boy with Intellectual Disability, Speech Delay and Mild Dysmorphic Features

**DOI:** 10.3390/genes11060707

**Published:** 2020-06-26

**Authors:** Orazio Palumbo, Pietro Palumbo, Ester Di Muro, Luigia Cinque, Antonio Petracca, Massimo Carella, Marco Castori

**Affiliations:** Division of Medical Genetics, Fondazione IRCCS-Casa Sollievo della Sofferenza, San Giovanni Rotondo, 71013 Foggia, Italy; p.palumbo@operapadrepio.it (P.P.); e.dimuro@operapadrepio.it (E.D.M.); l.cinque@operapadrepio.it (L.C.); a.petracca@operapadrepio.it (A.P.); m.carella@operapadrepio.it (M.C.); m.castori@operapadrepio.it (M.C.)

**Keywords:** 16q24.2q24.3 microduplication, high resolution SNP-Array analysis, emerging syndrome, neurodevelopmental disorders

## Abstract

No data on interstitial microduplications of the 16q24.2q24.3 chromosome region are available in the medical literature and remain extraordinarily rare in public databases. Here, we describe a boy with a *de novo* 16q24.2q24.3 microduplication at the Single Nucleotide Polymorphism (SNP)-array analysis spanning ~2.2 Mb and encompassing 38 genes. The patient showed mild-to-moderate intellectual disability, speech delay and mild dysmorphic features. In DECIPHER, we found six individuals carrying a “pure” overlapping microduplication. Although available data are very limited, genomic and phenotype comparison of our and previously annotated patients suggested a potential clinical relevance for 16q24.2q24.3 microduplication with a variable and not (yet) recognizable phenotype predominantly affecting cognition. Comparing the cytogenomic data of available individuals allowed us to delineate the smallest region of overlap involving 14 genes. Accordingly, we propose *ANKRD11*, *CDH15*, and *CTU2* as candidate genes for explaining the related neurodevelopmental manifestations shared by these patients. To the best of our knowledge, this is the first time that a clinical and molecular comparison among patients with overlapping 16q24.2q24.3 microduplication has been done. This study broadens our knowledge of the phenotypic consequences of 16q24.2q24.3 microduplication, providing supporting evidence of an emerging syndrome.

## 1. Introduction

Chromosome microarray analysis (CMA) has become a routine diagnostic test for autism spectrum disorders (ASD), intellectual disability (ID), developmental delay (DD), and multiple congenital anomalies (MCA), with a diagnostic yield of up to 15–20% in these cases [1,2,3,4]. In the last decade, this approach allowed the clinical and molecular characterization of an increasing number of microdeletion/microduplication syndromes which run unrecognized during standard cytogenetic analysis [5,6,7,8]. Nevertheless, the clinical interpretation of many CNVs remains challenging in a significant number of cases presenting rare or private rearrangements. In these circumstances, reporting new patients and case series and contributing to the growing knowledge of public databases is crucial for sharing difficulties and reaching collective solutions. Among these (only apparently) not recurrent rearrangements, 16q24.2q24.3 microduplication has never been reported in the medical literature and only a few patients with “pure” 16q24.2q24.3 microduplication have been submitted to public databases. Therefore, its clinical impact remains uncertain and the associated phenotypes poorly characterized. In this paper, we describe a 9-year-old boy, referred to us for a neurodevelopmental disorder, carrier of a *de novo* interstitial 16q24.2q24.3 microduplication spanning 2.2 Mb. The comparison of his clinical phenotype with that of other subjects previously annotated in DECIPHER and carrying overlapping duplications allowed us to propose a minimal set of shared features. In addition, we were able to define the smallest region of overlap (SRO) among patients, suggesting candidate genes for the observed clinical manifestations. Although the clinical relevance of this CNV remains to be refined by further studies, we suggest that the 16q24.2q24.3 microduplication may represent a potential novel syndromic form of neurodevelopmental disorder.

## 2. Materials and Methods

### 2.1. Genomic DNA Extraction and Quantification

This family gave their signed informed consent to molecular testing and to the full content of this publication. This study was in line with the 1984 Helsinki declaration and its subsequent revisions. Molecular testing carried out in this study is based on the routine clinical care performed at our Institute. Peripheral blood samples were taken from the patient and their parents, and genomic DNA was isolated by using Bio Robot EZ1 (Quiagen, Solna, Sweden). The quality of DNA was tested on 1% electrophorese agarose gel, and the concentrations were quantified with Nanodrop 2000 C spectrophotometer (Thermo Fisher Scientific, Waltham, MA, USA).

### 2.2. SNP-Array Analysis

High resolution SNP-array analysis of the proband and his parents was carried out by using the CytoScan HD array (Thermo Fisher Scientific, Waltham, MA, USA) as previously described [9]. Data analysis was performed using the Chromosome Analysis Suite Software version 4.1 (Thermo Fisher Scientific, Waltham, MA, USA) following a standardized pipeline. Briefly: (i) the raw data file (CEL) was normalized using the default options; (ii) an unpaired analysis was performed using as baseline 270 HapMap samples in order to obtain copy numbers value, while the amplified and/or deleted regions was detected using a standard Hidden Markov Model (HMM) method. We retained copy number variations (CNVs) ≥15 Kb in length and overlapping ≥10 consecutive probes to reduce the detection of false-positive calls. The significance of each detected CNV was determined by comparing all chromosomal alterations identified in the patient with those collected in an internal database of ~4500 patients studied by SNP arrays since 2010, and public databases including Database of Genomic Variants (DGV), DECIPHER, and ClinVar. Base pair positions, information about genomic regions and genes affected by CNVs, and known associated disease have been derived from the University of California Santa Cruz (UCSC) Genome Browser, build GRCh37 (hg19). The clinical significance of each rearrangements detected has been assessed following the American College of Medical Genetics (ACMG) guidelines [10].

### 2.3. Real Time Quantitative PCR

Specific target sequences were selected for Real-time quantitative PCR (qPCR) using Primer Express Software v3.0 (Thermo Fisher Scientific, Waltham, MA, USA) (*CDH15*/NM_004933, exon 6, Forward: GCAGGTGGCGGACATGTC; *CDH15*/NM_004933, exon 6, Reverse: GGGCATTGTCATTGATGTCATC. *MAP1LC3B*/NM_022818, exon 3, Forward: GAACGATACAAGGGTGAGAAGCA; *MAP1LC3B*/NM_022818, exon 3, Reverse: GACATGGTCAGGTACAAGGAACTTT). The qPCR was performed using Power SYBR Green PCR Master Mix (Thermo Fisher Scientific, Waltham, MA, USA). PCRs were run in triplicate on ABI PRISM 7900HT Sequence Detection System (Thermo Fisher Scientific, Waltham, MA, USA) and Cycling conditions were as follows: 2 min at 50 °C, 95 °C for 10 min, 40 cycles at 95 °C for 15 s and 60 °C for 1 min. Calculation of the gene copy number was made using the 2^−ΔΔCT^ method. Glyceraldehyde phosphate dehydrogenase (*GAPDH*), a described reference with a normal copy number, was chosen as housekeeping gene to normalize the related amount of target genes. Using this method, a Diploid Copy Number of 1 ± 0.2 is expected for a normal sample and a value of 1.5 ± 0.2 for a sample with duplication.

## 3. Results

### 3.1. Clinical Description

This is a 9-year-old boy, only child of unaffected and unrelated parents. He was born at term after an uneventful pregnancy with a birth weight of 3100 g (25th centile) and length of 49 cm (50th centile). Apgar score and head circumference at birth were not available. Neonatal period run unremarkable. Lactation, nutrition, and dentition were normal. The patient sat at six months, walked alone at 15 months, said his first words at 36 months, and gained full sphincter control at five years. Communicative skills improved short after the beginning of speech therapy, which was integrated by cognitive and physical therapy, and followed by a special educational program. At six years, the IQ was 56 (ID of mild degree) at the Leiter-R short scale, while the Childhood Autism Rating Scale 2 had a value of 28.5 not indicative of ASD. Paroxysmal spikes were noted at the electroencephalogram multiple times, but the patient never experienced seizures. Brain magnetic resonance imaging resulted normal. In addition, he had normal heart anatomy and function as assessed by echocardiogram; abdominal ultrasound examination, auditory evoked potential and eye exams were all normal. During physical examination, the patient did not display any significant facial dysmorphism, except for narrow and sloping forehead, bulbous nose with slightly anteverted nares. There are mild pectus excavatum involving the superior half of the sternum, pronounced fingerpads and bilateral clinodactyly of the fifth finger. External genitalia were normal. No additional anomalies were noted.

### 3.2. Molecular Findings

SNP-array analysis of the patient revealed microduplication involving the 16q24.2q24.3 chromosome region. The duplicated region was 2.2 Mb in size and covered by 1664 SNP array probes. Carrier testing in the parents, performed by chromosome microarrays analysis (CMA) using the same platform (i.e., CytoScan HD Array), resulted in normal outcomes indicating a de novo origin of the 16q24.2q24.3 microduplication in the patient (Figure 1).

Apart from known polymorphisms, no other CNVs were detected. qPCR performed on the patient and his parents confirmed the duplication in the patient and the lack of copy number change in the parents (data not shown). The molecular karyotype of the patient, according with the International System for Human Cytogenetic Nomenclature (ISCN 2016), is: arr(GrCh37) 16q24.2q24.3(87489142x2,87502161_89688617x3,89688904x2)dn.

The duplicated region in 16q24.2q24.3 contains 38 RefSeq genes (*ZCCHC14, JPH3, KLHDC4, SLC7A5, CA5A, BANP, ZNF469, ZFPM1, MIR5189, ZC3H18, IL17C, CYBA, MVD, MGC23284, SNAI3, RNF166, CTU2, PIEZO1, MIR4722, CDT1, APRT, GALNS, TRAPPC2L, PABPN1L, CBFA2T3, ACSF3, LINC00304, LOC400558, CDH15, SLC22A31, ZNF778, ANKRD11, LOC100287036, SPG7, RPL13, SNORD68, CPNE7, DPEP1*).

Consultation of the DGV did not reveal this region as a benign copy variable region. In ClinVar database we did not find any annotated patients with similar rearrangements, while, in DECIPHER, we found 14 cases with a microduplication in 16q24.2q24.3 overlapping with the one identified in our patient. Only six of them were reported to have a single comparable copy number gain (“pure” 16q24.2q24.3 microduplication).

## 4. Discussion

Here, we described a boy with ID, speech delay and mild facial dysmorphism, carrier of a de novo 16q24.2q24.3 microduplication identified by high-resolution SNP-array analysis. This rearrangement encompassed 38 RefSeq genes including nine OMIM genes (*JPH3, ZNF469, CYBA, CDT1, APRT, GALNS, CDH15, ANKRD11,* and *SPG7*).

To further investigate genotype-phenotype correlations, we searched for additional subjects carrying overlapping microduplications in PubMed and public databases. We found 14 cases with a microduplication in 16q24.2q24.3 overlapping with the identified one, but in only six patients (DECIPHER ID: 275865, 300593, 271478, 333548, 392985, 322843) the microduplication was not associated with other (potential) disease-causing rearrangements. In Table 1 are listed and compared the clinical and molecular findings in the six previously annotated and present patients, while a molecular comparison among them is shown in Figure 2A.

Interestingly, in all cases (four out of seven), including ours, in which parental DNA analysis was performed, the rearrangement occurred de novo. Although family segregation data was not available in three cases, this information together with the mean extension of the rearrangements, their significant overlap and genes content (see below) prompted us to attribute a “likely pathogenic” interpretation of the molecular finding in our patient.

From a clinical perspective, global developmental delay/ID is the most common finding (five in seven). Among the other neurodevelopmental attributes, seizures occurred in two and delayed speech and behavioral problems in one each. Somatic manifestations are diverse and included minor hands and feet abnormalities documented in two cases (i.e., pronounced fingerpads, and bilateral clinodactyly of the fifth finger in our patient; deep palmar crease and finger clinodactyly in DECIPHER 392985), congenital diaphragmatic hernia in one and multiple congenital anomalies, comprising abnormal septum pellucidum, dextrocardia, anal fistula and abnormality of the labia, in one. Concerning the impact of these manifestations, the paucity of available data (i.e., for patient DECIPHER 300593 and 271478 clinical information is lacking) and the extreme heterogeneity of age at ascertainment (i.e., two subjects were less than 1 year of age) significantly limit pattern recognition. Therefore, we think that some of the clinical features that seem to be not frequent, such as seizures, delayed speech/language development, and behavioral problems, could be simply underestimated or never systematically investigated and reported to date. For this reason, we suggest a regular neurobehavioral evaluation for patients carrying a 16q24.2q24.3 microduplication in order to characterize more in detail the clinical features of this emerging disorder.

Finally, facial dysmorphisms are reported in three in seven cases. In detail, two individuals (i.e., DECIPHER 392985 and ours) shared narrow and sloping forehead and bulbous nose, while “abnormal facial shape” was simply reported in the remaining one (DECIPHER 275865). Assessing facial features is further complicated by the variable age at clinical evaluation. Therefore, it is currently not possible to delineate a recurrent pattern also for the somatic features.

Delineation of a distinctive pattern of clinical manifestations in individuals with 16q24.2q24.3 microduplication could be useful to suggest a clinical diagnosis, to speed up the diagnostic process improving, if possible, the patient care and management. We suggest describing in the literature or including in public database such as DECIPHER detailed clinical information about individuals with a 16q24.2q24.3 microduplication.

From a molecular perspective, the comparison of the duplicated chromosome region among the different patients allowed us to identify the SRO in a 724 Kb segment with the proximal (centromeric) breakpoint at 88,755,341 bp, found in DECIPHER patient 300593, and the distal (telomeric) breakpoint at 89,479,707 bp, found in DECIPHER patient 275865 (Figure 2A). The SRO contains 14 genes (Figure 2B). Among these genes, we propose *ANKRD11*, *CDH15*, and *CUT2* as the most possible candidates for contributing to the etiology of the neurodevelopmental manifestations shared by the patients.

*ANKRD11* encodes the ankyrin repeat domain containing protein 11 and its haploinsufficiency, resulting from either loss-of-function variants, 16q24.3 microdeletions or intragenic microduplications, has been documented in patients with the KBG syndrome (OMIM #148050), a rare developmental disorder characterized by ID, ASD and distinctive craniofacial features [11,12,13,14]. *ANKRD11* gene deletion has also been reported in ASD and variable cognitive impairment in the absence of a syndromic presentation [15] as well as in subjects with less specific *KBG-like* phenotypes [16]. Therefore, as suggested by other authors [17], it is reasonable to assume that *ANKRD11* is dose-sensitive and that it may affect development also in case of overexpression, functional mechanism presumed in the case of 16q24.2q24.3 microduplication. Finally, in support of the suggestive implication of *ANKRD11* gene as candidate, there are several examples of well characterized syndromic conditions for which both point mutations in single gene as well as CNVs involving that gene are known to be causative of clinical phenotype (i.e., *RAI1*/17p11.2, Smith-Magenis syndrome, and *SHANK3*/22q13, Phelan–McDermid syndrome) [18,19].

*CDH15* encodes a calcium dependent cell adhesion molecule belonging to the cadherin family (cadherin 15). Cadherins are transmembrane glycoproteins consisting of an extracellular domain, a transmembrane region and a cytoplasmic domain. The extracellular domains mediate Ca^2+^-dependent intercellular adhesion by homophilic interactions. The binding properties and specificities of the adhesive function are located in the N-terminal part of the molecules [20]. Heterozygous variants in *CDH15* have been reported in families with autosomal dominant intellectual disability type 3 (MRD3; OMIM #612580). Also, in vitro functional studies showed that mutant proteins result in decreased cell adhesion suggesting that *CDH15* alterations, either alone or in combination with other factors, likely play a role in the etiology of ID [21]. Finally, copy number variations (both deletions and duplications) affecting other genes involved in neural cell adhesion molecules have been recently associated with neurodevelopmental disorders [22,23]. Accordingly, 16q24.2q24.3 microduplication can be added to available data corroborating a key role of these cellular pathways in cognitive development.

*CTU2* is an additional candidate gene mapping into 16q24.2q24.3 microduplication SRO and encoding a protein involved in the post-transcriptional modification of transfer RNAs (tRNAs). This protein plays a role in thiolation of uridine residue present at the wobble position in a subset of tRNAs, resulting in enhanced codon reading accuracy. Biallelic variants in *CTU2* have been associated with a specific syndromic phenotype featuring microcephaly, facial dysmorphism, renal agenesis, and ambiguous genitalia [24,25], and this gene has been recently listed into the Developmental Disorders Genotype-Phenotype Database (DDG2P).

Altogether, the evidence emerging from our study and the current knowledge concerning the proposed candidate genes support our hypothesis that their copy number alteration contribute to the etiology of the clinical phenotype observed in patients with 16q24.2q24.3 microduplication mainly for neurodevelopmental features shared among affected individuals.

For the other genes duplicated in patients discussed in the present study, although none of them seem to be clearly associable with the clinical traits reported, we cannot exclude their involvement in the etiology of the clinical condition. More detailed genetic and/or functional studies, or patients with point mutations/CNVs affecting only one or a few of these genes, are needed to elucidate this possibility.

## 5. Conclusions

Here, we reported a nine-year-old boy with a pure de novo 16q24.2q24.3 microduplication, a molecular finding which was previously unreported. Scrutiny of available databases allowed the identification of six additional subjects with similar genotypes. The careful analysis of data, carried out comparing the available patients both from a clinical than from a molecular point of view, suggest clinical relevance for this CNV providing supportive evidence of an emerging syndrome. In addition, we identified the SRO of 724 Kb involving 14 genes. Among them, on the basis of functional and clinical data from the medical literature, we proposed *ANKRD11*, *CDH15* and *CTU2* as best candidates for explaining the neurodevelopmental manifestations of 16q24.2q24.3 microduplication. Obviously, the publication of additional patients and their submission to public databases, further functional studies or animal models are needed to corroborate our hypothesis, to establish a more accurate genotype–phenotype correlation and to verify the existence of an associated recurrent phenotype.

## Figures and Tables

**Figure 1 genes-11-00707-f001:**
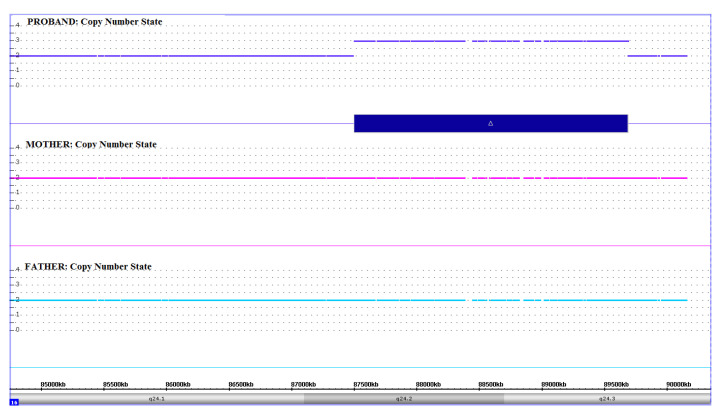
Results of SNP-Array analysis in the patient and his parents. Copy number state of each probe is drawn along chromosome 16 from 85 to 90 Mb (UCSC Genome Browser, build GRCh37/hg19). The upper panel represents the copy number state of the proband, the middle panel that of the mother and the lower panel that of the father. Values of Y-axis indicate the inferred copy number according the probes intensities. Blue bar indicates the duplication identified in the patient.

**Figure 2 genes-11-00707-f002:**
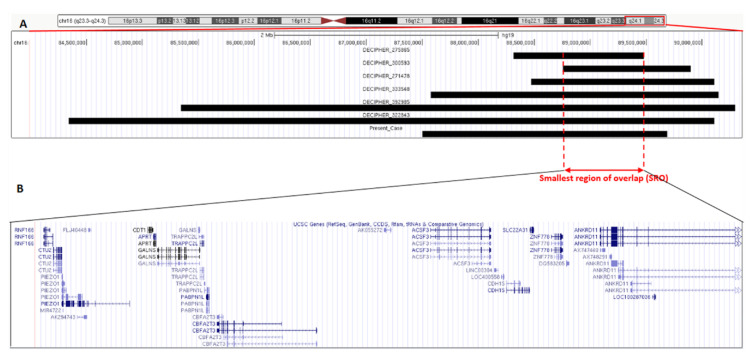
Schematic representation of chromosome 16q24 from 84 to 90,5 Mb (assembly GRCh37/hg19) indicating (**A**) the duplicated region in our patient and in the patients from DECIPHER (black bars). Red vertical dashed lines delimitate the smallest regions of overlap (SRO) among all patients. (**B**) Genes included in the SRO (*RNF166, CTU2, PIEZO1, CDT1, APRT, GALNS, TRAPPC2L, PABPN1L, CBFA2T3, ACSF3, CDH15, SLC22A31, ZNF778, ANKRD11*).

**Table 1 genes-11-00707-t001:** Clinical and molecular features of patients with overlapping 16q24.2q24.3 microduplication.

	Present Case	Decipher 275865	Decipher 300593	Decipher 271478	Decipher 333548	Decipher 392985	Decipher 322843
**Age at Last Clinical Assessment**	9 years	<1 year	4 years	8 years	7 years	<1 year	8 years
**Gender**	M	F	M	M	F	F	M
**Chromosome**	16	16	16	16	16	16	16
**Cytoband**	q24.2q24.3	q24.2q24.3	q24.3	q24.2q24.3	q24.2q24.3	q24.1q24.3	q24.1q24.3
**Type**	Gain	Gain	Gain	Gain	Gain	Gain	Gain
**Start (hg19, bp)**	87502161	88317010	88755341	88473369	87577020	85342500	84341219
**Stop (hg19, bp)**	89688617	89479707	89897010	90111263	90148393	90294753	90111263
**Size (Mb)**	2.19	1.16	1.14	1.64	2.57	4.95	5.77
**# genes**	38	25	32	51	57	81	90
**Inheritance**	*dn*	*dn*	ND	ND	ND	*dn*	*dn*
**Clinical Significance**	LP	LP	LP	VUS	VUS	LP	P
**Global Developmental Delay/Intellectual Disability**	+	+	NR	NR	+	+	+
**Delayed Speech/Language development**	+	ND	NR	NR	-	ND	-
**Behavioral Problems**	-	ND	NR	NR	-	ND	+
**Seizures**	-	+	NR	NR	-	-	+
**Dysmorphic Features**	Narrow and sloping forehead, bulbous nose with slightly anteverted nares	Abnormal facial shape	NR	NR	-	High anterior hairline, narrow and sloping forehead, bulbous nose, prominent nasal bridge, aplasia/Hypoplasia of the earlobes, hypertelorism, Micrognathia	-
**Hands and Feet Abnormalities**	Pronounced fingerpads, bilateral clinodactyly of the fifth finger	-	NR	NR	-	Deep palmar crease, finger clinodactyly	-
**Others congenital abnormalities**	CDH	-	NR	NR	-	MCA	-

M, male; *dn*, de novo; LP, likely pathogenetic; +, feature present; -, feature absent; CDH, congenital diaphragmatic hernia; F, female; ND, not determined; NR, not reported; VUS, variant of unknown significance; MCA, multiple congenital anomalies; P, pathogenetic.

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
