# Peer review of "A Private 16q24.2q24.3 Microduplication in a Boy with Intellectual Disability, Speech Delay and Mild Dysmorphic Features"

_genes, 2020, doi:10.3390/genes11060707_

Round 1
Reviewer 1 Report
Palumbo and colleagues describe an interesting case report of a patient carrying an interstitial microduplication at 16q24.2q24.3. In addition, the authors provide a well-done phenotypical and molecular comparison of the reported patient with other cases of microduplication reported on public databases.
However, the authors should extend their comparison to the patients described by Crippa and colleagues in 2015 (https://pubmed.ncbi.nlm.nih.gov/25838844/). In this paper, the authors describe three individuals carrying a 89 kb microduplication at 16q24.3 affecting ANKRD11 only.
Since another microduplication case involving the same region has already been reported, the authors should tune down references to the novelty of their reported microduplication. Also, they should re-interpret the smallest region of overlap by considering the patients of Crippa et al.
Author Response
Author's Reply to the Review Report (Reviewer 1)
Palumbo and colleagues describe an interesting case report of a patient carrying an interstitial microduplication at 16q24.2q24.3. In addition, the authors provide a well-done phenotypical and molecular comparison of the reported patient with other cases of microduplication reported on public databases.
However, the authors should extend their comparison to the patients described by Crippa and colleagues in 2015 (https://pubmed.ncbi.nlm.nih.gov/25838844/). In this paper, the authors describe three individuals carrying a 89 kb microduplication at 16q24.3 affecting ANKRD11 only. Since another microduplication case involving the same region has already been reported, the authors should tune down references to the novelty of their reported microduplication. Also, they should re-interpret the smallest region of overlap by considering the patients of Crippa et al.
Response 1: we know very well the excellent paper written by Crippa et al., but we did not include their patients in the clinical/molecular comparison carried out in our study for several molecular, functional end clinical reasons.
In detail, we excluded them from the comparison performed because: i) they carried a rearrangement too small compared to the one identified in the other seven cases (in selecting of patients for clinical and molecular comparison one of the inclusion criteria was that they were carriers of a microduplication similar in genomic location and size to that identified in our patient); ii) Crippa and colleagues in 2015 identified an intragenic microduplication extends from ANKRD11 intron 2 to exon 9 (or intron 10, depending on the minimum and maximum size of the duplication inferred from reference transcript NM_013275.5), which has as functional consequence the ANKRD11 haploinsufficiency and, similarly to deletions and point mutations, results in KBG syndrome (in our patient and in the selected cases from DECIPHER the duplication include the entire ANKRD11 gene and in this case haploinsufficiency is unlikely); iii) Crippa’s patients were affected by KBG syndrome (while the cases selected for our paper no) due to the haploinsufficiency of ANKRD11.
The main topic of our paper was not the KBG and the ANKRD11 haploinsufficiency but its opposite that is a probable new emerging NDD syndrome caused by the presumed overexpression of some genes including ANKRD11. We performed this clinical and molecular comparison to suggest that microduplication of 16q24.2q24.3 chromosomal region may represents an emerging syndrome. Obviously, additional patients with overlapping genomic rearrangements are needed to corroborate our hypothesis.
Anyway, since we believe that Crippa's paper has contributed considerably to the understanding of the molecular mechanisms underlying KBG expanding our knowledge on the functional consequences of CNVs affecting the ANKRD11 gene, we included the article in the reference list (as 14) and re-discussed part of the discussion about the ANKRD11 gene as candidate. In detail, we state: “ANKRD11 encodes the ankyrin repeat domain containing protein 11 and its haploinsufficiency, resulting from either loss-of-function variants, 16q24.3 microdeletions or intragenic microduplication, has been documented in patients with the KBG syndrome (OMIM # 148050) , a rare developmental disorder characterized by ID, ASD and distinctive craniofacial features [11-14]”

Reviewer 2 Report
This peer-review was completed with the help of Dr. Kuntal Sen, child neurology fellow at Children’s National Hospital.
In this paper, Palumbo et al. report a patient with developmental delay and dysmorphic features as a consequence of 16q24.2q24.3 microduplication and propose candidate genes for this neurodevelopmental disorder. This is a rare microduplication syndrome with vey limited reports and therefore, this case certainly contributes to literature. Overall, the manuscript is very well-written and easy to follow.
Comments/Questions:
- What was the reason to order EEG – were there any seizure-like episodes? If yes, please provide description. Please include information about hearing and eye exam, echo, Abdominal US if available which are a part of regular syndromic surveillance.
- Please clarify if carrier testing in parents was using CMA or FISH?
- The phenotypic overlap with previously reported patients is significant and de novo nature of the duplication is suspicious. However exome sequencing or atleast an intellectual disability panel to exclude other potential genetic variants would have been crucial to annotate this particular duplication as pathogenic for the patient.
- In table 1, please include MRI/ neuroimaging findings in other patients if available.
- The proposed implication of ANKRD11 gene is interesting. Indeed, KBG syndrome has a neurodevelopmental phenotype in addition to other ectodermal defects. Single nucleotide variants in single gene as well as microdeletions involving that gene causing same phenotype is a well-understood phenomenon in clinical genetics (for eg, RAI1/ 1711.2- Smith-Magenis syndrome, SHANK3/ 22q13- Phelan-McDermid syndrome). This might be worth discussing to support the candidacy of ANKRD11.
Author Response
Author's Reply to the Review Report (Reviewer 2)
In this paper, Palumbo et al. report a patient with developmental delay and dysmorphic features as a consequence of 16q24.2q24.3 microduplication and propose candidate genes for this neurodevelopmental disorder. This is a rare microduplication syndrome with very limited reports and therefore, this case certainly contributes to literature. Overall, the manuscript is very well-written and easy to follow.
Comments/Questions:
1. What was the reason to order EEG – were there any seizure-like episodes? If yes, please provide description. Please include information about hearing and eye exam, echo, Abdominal US if available which are a part of regular syndromic surveillance.
Response 1: the patient underwent EEG as this investigation is one of those practiced in the diagnostic routine in childhood neuropsychiatry for patients with psychomotor retardation, intellectual disability, autism spectrum disorders of unknown etiology. There were not any seizure-like episodes and the patient never experienced seizures, as stated in the section 3.1 Clinical Description.
In agreement with reviewer, we included information about hearing and eye exam, echo. In detail, in the section 3.1 Clinical Description we added: “In addition, he had normal heart anatomy and function as assessed by echocardiogram; abdominal ultrasound examination, auditory evoked potential and eye exams were all normal.”
2. Please clarify if carrier testing in parents was using CMA or FISH?
Response 2: carrier testing was performed by using CMA. In the section 3.2 Molecular findings we stated that carrier testing in the parents using the same platform resulted in…. that is high resolution SNP-Array analysis.
In addition, in the Legend of Figure 1 we clearly stated “Results of SNP-Array analysis in the patient and his parents” indicating that it was performed using CMA.
Anyway, to be more explicative and clear, we change the period in: “SNP-array analysis of the patient revealed microduplication involving the 16q24.2q24.3 chromosome region….Carrier testing in the parents, performed by chromosome microarrays analysis (CMA) using the same platform (i.e. CytoScan HD Array), resulted in normal outcomes indicating a de novo origin of the 16q24.2q24.3 microduplication in the patient”.
3. The phenotypic overlap with previously reported patients is significant and de novo nature of the duplication is suspicious. However exome sequencing or at least an intellectual disability panel to exclude other potential genetic variants would have been crucial to annotate this particular duplication as pathogenic for the patient.
Response 3: in the meantime, since we submitted the paper for the first time, to validate the role of the 16q24.2q24.3 microduplication identified in the patient, we analyzed a panel containing about 100 NDD-related genes (ANKRD11, ARX, ASXL1, ASXL2, ASXL3, ATAD3A, AUTS2, BRPF1, CASK, CDH15, CDH5, CDH8, CHL1, CNTN4, CNTN5, CNTN6, CNTNAP2, CTNNB1, DDX3X, DHCR7, DLG3, DOCK8, DYRK1A, EBF3, EHMT1, FGD1, FOXP1, FOXP2, GATAD2B, GDI1, GRIA3, GRIK2, GRIN2A, K1, GRA2, JAT1, K1 KATNAL1, KDM5C, KIAA2022, KIRREL3, MBD5, MECP2, MED12, MED13L, MED23, MEF2C, MID1, NACC1, NLGN3, NLGN4X, NRXN1, OPHN1, OTUD6B, PAK3, PCDH10, PTD, PSDH19, PPDH19, PPM RAI1, RERE, SATB2, SETBP1, SETD5, SHANK3, SLC6A8, SLC9A6, SON, SRPX2, STAG1, STAG2, SYNGAP1, SYP, TAF1, TBCK, TBL1XR1, TBR1, TCF4, TECR, TRIO, TRIP12, TSPAN, TSPAN, TSPAN USP9X, YWHAE, YY1, ZDHHC9, ZEB2). Following this approach, we did not identified any variant classifiable as pathogenetic, likely pathogenetic or VOUS (variant of unknown clinical significance), according to the American College of Medical Genetics (ACMG) guidelines (Genet. Med., 2011, 13, 680-685).
4. In table 1, please include MRI/ neuroimaging findings in other patients if available.
Response 4: We did not include any MRI/neuroimaging findings for other patients because they were not available.
5. The proposed implication of ANKRD11 gene is interesting. Indeed, KBG syndrome has a neurodevelopmental phenotype in addition to other ectodermal defects. Single nucleotide variants in single gene as well as microdeletions involving that gene causing same phenotype is a well-understood phenomenon in clinical genetics (for eg, RAI1/ 17q11.2- Smith-Magenis syndrome, SHANK3/ 22q13- Phelan-McDermid syndrome). This might be worth discussing to support the candidacy of ANKRD11.
Response 5: in agreement with the reviewer, to support the candidacy of ANKRD11, we added a brief discussion on this well-understood phenomenon in clinical genetics that is single nucleotide variants in single gene as well as copy number variants involving that gene causing clinical phenotype.
In detail, we added the sentence: “Finally, in support of the suggestive implication of ANKRD11 gene as candidate, there are several examples of well characterized syndromic conditions for which both point mutations in single gene as well as CNVs involving that gene are known to be causative of clinical phenotype (i.e. RAI1/17p11.2-Smith-Magenis syndrome, SHANK3/ 22q13- Phelan-McDermid syndrome) [18, 19].”
In addition, we mentioned the suggested syndromes (i.e. Smith-Magenis syndrome and Phelan-McDermid syndrome), added proper references (i.e 18 and 19) and adjusted the references list that now include 25 cited articles.
